# Pfizer-BioNTech (BNT162b2) Vaccine Effectiveness against Symptomatic Laboratory-Confirmed COVID-19 Infection among Outpatients in Sentinel Sites, Lebanon, July–December 2021

**DOI:** 10.3390/vaccines12090954

**Published:** 2024-08-23

**Authors:** Lina Chaito, Pawel Stefanoff, Joaquin Baruch, Zeina Farah, Mona Albuaini, Nada Ghosn

**Affiliations:** 1Epidemiological Surveillance Program, Ministry of Public Health, Beirut 0127, Lebanon; 2Mediterranean and Black Sea Programme in Intervention Epidemiology Training (MediPIET), European Centre for Disease Prevention and Control (ECDC), 171 83 Stockholm, Sweden; 3European Programme for Intervention Epidemiology Training Program (EPIET), European Centre for Disease Prevention and Control (ECDC), 171 83 Stockholm, Sweden; 4National Influenza Center (NIC), Rafic Hariri University Hospital (RHUH), Beirut 0127, Lebanon

**Keywords:** COVID-19, vaccine, effectiveness, symptomatic outcome

## Abstract

On 14 February 2021, Lebanon implemented nationwide vaccination, offering the Pfizer-BioNTech (BNT162b2) vaccine to adults over 50 years of age. We estimated the effectiveness of the Pfizer-BioNTech vaccine in preventing symptomatic laboratory-confirmed COVID-19. We conducted a test-negative case–control (TND) study among symptomatic adults aged 50 years and older who presented with influenza-like illness (ILI) or COVID-19-like illness (CLI) in surveillance sentinel sites between 1 July and 31 December 2021. Unvaccinated participants did not receive any vaccine dose before symptom onset. Vaccinated participants received at least one dose within 14 days before onset of symptoms. We estimated vaccine effectiveness against symptomatic laboratory-confirmed COVID-19, adjusted for demographic and behavioral factors, using multivariable logistic regression. Out of 457 participants with symptoms, 150 (33%) were positive and 307 (67%) were negative for SARS-CoV-2. Adjusted vaccine effectiveness was 22% (95% CI: −70–65%) for those partially vaccinated and 44% (95% CI: 6–67%) for those fully vaccinated. Vaccination with two doses of the Pfizer-BioNTech vaccine was effective in preventing COVID-19 symptomatic illness in the older population. Vaccine effectiveness was lower for those partially vaccinated. We recommend enhancing vaccine uptake with at least one dose among risk groups for COVID-19 and keeping general recommendations on contact and droplet precautions in the general population.

## 1. Introduction

On 14 February 2021, Lebanon started the COVID-19 national vaccination campaign. Vaccination was deployed in four phases based on age, profession, and underlying conditions. Four vaccine brands were used in Lebanon: Pfizer-BioNTech, Oxford–AstraZeneca, Sputnik V and Sinopharm [1].

These vaccines were developed using various technologies and have played a crucial role in controlling the pandemic. The Pfizer-BioNTech vaccine was developed using the innovative mRNA technology that enables rapid manufacturing, unlike traditional vaccines. This speed was crucial in responding to the changing SARS-2 virus strains during the pandemic. Oxford–AstraZeneca and Sputnik were based on non-replicating viral vector vaccine technology while Sinopharm was based on the traditional technology of inoculating the killed or inactivated virus. The widespread availability of these vaccines, especially in low- and middle-income countries, helped prevent countless cases of severe illness and death [2,3,4].

The delivery of COVID-19 vaccines in Lebanon initially occurred through public and private hospitals and then extended to include mass vaccination centers and primary healthcare centers in all Lebanese regions. Pfizer-BioNTech was the primary vaccine used for older adults, including those aged 50 years and above. Its efficacy and widespread global approval made it a cornerstone of Lebanon’s vaccination strategy [1].

In the first half of 2021, nationwide vaccination campaigns had led to a significant decrease in COVID-19 incidence and mortality. In June 2021, the country experienced a resurgence of COVID-19, dominated by the Delta (B.1.617.2) variant of SARS-CoV-2 [5]. Studies in various contexts reported different estimates of COVID-19 vaccines’ effectiveness against clinical outcomes of SARS-CoV-2 during the wave dominated by the Delta (B.1.617.2) variant [6,7].

Clinical trials had demonstrated the high efficacy of COVID-19 vaccines against hospitalization (93%; 95% CI: 83–97%), death (82%; 95% CI: 81–97%) and symptomatic disease (81%; 95% CI: 73–87%) [8]. However, vaccine effectiveness observed in real life conditions, such as that of a mass vaccination program, may differ and it is essential to document and monitor it after clinical trials. Real life conditions include regional differences in population characteristics, the presence of underlying conditions, cold chain maintenance, vaccine brands, dose interval and emerging variants of concern [9], which requires the continuous monitoring of vaccine effectiveness to adjust public health recommendations [10,11].

In this study, we aimed to estimate the effectiveness of partial and full vaccination with the Pfizer-BioNTech vaccine in preventing symptomatic laboratory-confirmed disease and to use results for guiding public health policies and recommendations to the general population.

## 2. Materials and Methods

### 2.1. Study Design

We conducted a test-negative case–control (TND) study among patients presenting with influenza-like illness (ILI) and COVID-19-like illness (CLI) at sentinel sites with a set of symptoms meeting the clinical case definition [12,13]. We selected all patients consulted at ILI/CLI sentinel sites between 1 July 2021 and 31 December 2021, for whom PCR testing results were available. Cases were patients who tested positive for SARS-CoV-2. Controls were patients who tested negative.

### 2.2. Study Population

The study population consisted of persons consulting at sentinel outpatient clinics with upper respiratory infection symptoms (referred to as ILI or CLI) during the study period. One day per week, the first 22 symptomatic outpatients visiting each one of the clinics participating in the national ILI sentinel system were recruited for testing.

### 2.3. Inclusion and Exclusion Criteria

We included all consenting symptomatic adults aged ≥50 years tested for SARS-CoV-2 and eligible for vaccination. We excluded patients with unknown COVID-19 vaccination status, those vaccinated with COVID-19 vaccines other than the Pfizer-BioNTech vaccine, participants declaring a contraindication for the COVID-19 vaccine, unvaccinated with prior infection in the past 3 months and controls having symptoms compatible with COVID-19, such as dysgeusia and anosmia.

### 2.4. Data Collection

We accessed the sentinel sites database on 26 February 2022. We reviewed the database to extract demographic, clinical and laboratory information. We attempted contacting all eligible patients consulted for ILI/CLI symptoms during the study period. We conducted structured phone interviews with consenting eligible respondents. The interviews included information on medical care sought for ILI/CLI, vaccination status and the known confounders identified in the medical literature.

### 2.5. Outcome

The primary outcome of interest was PCR laboratory-confirmed COVID-19 symptomatic infection.

### 2.6. Exposure Assessment

We collected the information on COVID-19 vaccination during interviews and by requesting a copy of respondent vaccination cards by WhatsApp. Data on vaccination included the vaccine brand name, number of doses, dates of vaccination, adverse effects following immunization and reasons for not being vaccinated. We defined fully vaccinated as respondents who received two doses of the Pfizer-BioNTech vaccine, with the second dose received ≥14 days before the onset of symptoms. We defined partially vaccinated as respondents who received one dose of the Pfizer-BioNTech vaccine ≥14 days before the onset of symptoms or received two doses of the Pfizer-BioNTech vaccine, with the second dose received ≤14 days before the onset of symptoms. We defined unvaccinated as respondents who had not received the Pfizer-BioNTech vaccine before the onset of symptoms or were vaccinated with one dose ≤14 days before symptom onset.

### 2.7. Covariates

We collected data on multiple covariates that may be associated with the likelihood of being offered or accepting a vaccine and the risk of exposure to COVID-19 such as smoking, health status, living conditions, community exposures to COVID-19 and the practice of non-pharmaceutical interventions (see Appendix A).

### 2.8. Sampling Procedure

The sample size was estimated by referring to World Health Organization (WHO) interim guidance on the evaluation of COVID-19 vaccine effectiveness [14].

Considering the vaccination coverage (50%), effectiveness of the Pfizer-BioNTech vaccine (80%), ratio of controls to cases (2:1), precision of the estimate (±10%) and type 1 error rate (0.05), the minimum sample size for the study was calculated to be 153 cases and 306 controls. We inflated the sample size and included all the study population to account for non-participation, adjustment for effect modifiers and confounders, or exclusion factors that might be identified after enrolment. Thus, we accepted as sufficient the achieved sample of 227 cases and 527 controls.

### 2.9. Statistical Analysis

All data were analyzed using STATA (version 13) software. We initially compared the baseline characteristics between cases and controls. Univariate analysis was used to assess the distribution of covariates among cases and controls and to identify potential confounders. We used the Chi-squared test or Fisher’s exact test to compare categorical variables between cases and controls. We used Student’s t-test or Wilcoxon rank sum tests for continuous variables. The protective effect of the Pfizer-BioNTech vaccine was expressed as ratio of the odds of being vaccinated among cases versus the odds of being vaccinated among controls. The vaccine effectiveness (VE) was calculated using the following formula: (VE = (1 − OR) × 100%). We used a 95% confidence level for all statistical tests.

### 2.10. Ethical Considerations

The study was approved by the ethical task group for study review at the epidemiological surveillance unit of the Lebanese Ministry of Public Health. Informed oral consent was obtained from study subjects prior to participation and documented in the database. Data were de-identified by removing participants’ names and phone numbers after the completion of data collection.

## 3. Results

### 3.1. Summary of Enrolment

From 1 July until 31 December 2021, 754 outpatients visiting ILI/CLI sentinel sites and meeting inclusion criteria were tested for SARS-CoV-2. After excluding 297 patients, we included a total of 457 respondents with 150 cases (33%) and 307 controls (67%) (Figure 1).

### 3.2. Patient Characteristics

The general characteristics of respondents are shown in Table 1. There were 58% percent who were women, and 74% were Lebanese. The median (interquartile range, IQR) age was 59 (54–64) years. Most participants had at least one comorbidity (89 [59%] cases; vs 192 [63%] controls). There were more smokers among cases (82; 57%) than among controls (120; 39%) (*p* = 0.001). Regarding living conditions, the crowding index for cases and controls was 1.4 (±0.76) and 1.5 (±1.1), respectively.

### 3.3. Behavioral Factors

A comparison of behavioral factors between cases and controls showed no difference between the two groups in terms of exposure to SARS-CoV-2 in the community and the practice of non-pharmaceutical interventions (Appendix A).

### 3.4. COVID-19 Vaccination

A total of 231 (50%) participants were unvaccinated. Of the 226 (50%) vaccinated participants, 41 (9%) were partially vaccinated and 185 (41%) were fully vaccinated. More controls than cases were vaccinated with at least one dose (162 controls; 53% and 64 cases; 43%) (*p* = 0.04). For those vaccinated, the median delay between the administration of the first vaccine dose and the onset of disease symptoms was longer in controls (45 days) than in cases (25 days). The median delay between the administration of the second dose and the onset of disease symptoms was longer among cases (116 days) than among controls (98 days) (Table 2).

### 3.5. Vaccine Effectiveness against Laboratory-Confirmed SARS-CoV-2

Considering at least 14 days after vaccine administration, 45 cases (34%) and 116 controls (43%) were fully vaccinated and the corresponding vaccine effectiveness was 35% (95% CI: −2; 56). After adjusting for demographic variables, exposure to SARS-CoV-2 in the community, the practice of non-pharmaceutical interventions, VE was 44% (95% CI: 6; 67). A total of 11 cases (8%) and 25 controls (9%) were partially vaccinated and the corresponding VE was 26% (95% CI: −60; 66). After adjustment, multivariable analysis yielded a VE of 22% (95% CI: −70; 65) (Table 3).

## 4. Discussion

This study was conducted in Lebanon between July and December 2021. It provided real-world evidence for the effectiveness of the Pfizer-BioNTech vaccine against laboratory-confirmed symptomatic COVID-19 disease. We estimated that partial vaccination is 22% effective in preventing symptomatic disease in adults aged 50 years and older in Lebanon and that full vaccination is 44% effective.

Our VE estimates for symptomatic COVID-19 disease were lower than results reported by other studies. Findings from Canada and England indicated VE estimates of 57–70% among those partially vaccinated with a COVID-19 vaccine and 89–91% among those fully vaccinated [15,16]. Studies from Spain and Qatar reported lower VE point estimates among those partially vaccinated (30%) and fully vaccinated (77%) [17,18].

The lower VE estimates of our study compared to other studies might be explained by different factors. During the Alpha phase, vaccine effectiveness (VE) did not decline with increasing time since vaccination. However, in the Delta phase, there were significant reductions in VE as time from vaccination increased [19]. The studies mentioned above with a higher VE [15,16,17,18] all measured VE in the first weeks post-vaccination. In contrast, our study had a median time from vaccination to symptom onset of 116 days, explaining the lower VE results than other studies but consistent with a prolonged time from vaccination and the weaning of vaccine-derived immunity. Regarding the characteristics of our study population, which is composed of adults aged ≥50 years, the majority had comorbidities. Also, controls recruited in our study could have been previously infected before our study period by SARS-CoV-2 and therefore protected by prior undiagnosed or asymptomatic infections.

Despite the fact that our results indicated moderate vaccine effectiveness for two doses of the Pfizer-BioNTech vaccine, they underline the protective effect of vaccination among high-risk groups exposed to SARS-CoV-2. A study from the US indicated that vaccine breakthrough infections were found to be less infectious than primary infections in unvaccinated persons [20]. Also, it is important to emphasize that the literature to date consistently reports much higher vaccine effectiveness (more than 90%) against severe illness, hospitalization, and death [21]. As documented in a previous Lebanese vaccine evaluation study, the Pfizer-BioNTech vaccine has demonstrated effectiveness against severe COVID-19 disease: 93% (95% CI: 83–97%) for hospital admission, and 82% (95% CI: 81–97%) for fatal outcome [22].

The routine monitoring of vaccine effectiveness is crucial to inform public health policies and the general population. The routine monitoring of vaccine effectiveness can detect any change that could be due to an emerging variant of the virus or a waning in the vaccine protection.

In Lebanon, since the emergence of the COVID-19 pandemic, the established sentinel surveillance system to monitor influenza-like illness was also used to monitor COVID-19-like illness by the routine collection of respiratory specimens from symptomatic outpatients. The sentinel surveillance system can facilitate the conducting of various studies, including repetitive VE estimations based on an approved methodology [14]. The advantages of using ILI/CLI surveillance sentinels are that both cases and controls had sought medical care at the same facilities. Hence, they represent the same communities, reducing bias related to socio-economic status, vaccine access and disease risk. Also, this approach reduces confounding due to differences in healthcare-seeking behavior or access between cases and controls, which is often a source of bias. Sustaining the sentinel system can facilitate the conducting of repeated vaccine effectiveness studies, and thus provide updated data on vaccine protection for different age groups, vaccines and variants.

Our findings are subject to certain limitations. First, we could only determine the effectiveness of the Pfizer-BioNTech vaccine since the age group of our study population was only eligible for this vaccine brand. Second, study participants were sampled from the ILI/CLI database, and patients who seek healthcare from these facilities usually belong to deprived or poor segments of the population. Thus, the generalizability of our findings to the broader target population in Lebanon is uncertain. Third, restricting the age of the study population led to a smaller sample size and wider confidence intervals for the outcome estimates. Moreover, the initial sample size decreased due to a high rate of not-answered calls and a refusal to participate. Our sample size was not sufficient to conduct sub-group analysis to estimate the vaccine effectiveness over time since vaccination. In addition, as in any observational study, unmeasured confounding might exist. Lastly, some information was self-reported, introducing recall errors and missing data; however, this was not the case for our outcome (COVID-19 test) and main exposure (vaccination status by WhatsApp photo) data and therefore not likely to affect the main findings of the study.

## 5. Conclusions

We estimated a moderate effectiveness of two doses of the Pfizer-BioNTech vaccine against symptomatic laboratory-confirmed symptomatic COVID-19 illness among an adult population aged 50 years and above. The vaccine was less effective among partially vaccinated adults. We recommend increasing efforts to enhance vaccine uptake with at least one dose of the COVID-19 vaccine among risk groups. In addition, we recommend general contact and droplet precautions in the general population during the periods of high SARS-CoV-2 circulation. In the longer term, we recommend conducting periodic vaccine effectiveness studies regarding age categories, risk groups and time since vaccination. Test-negative studies in the ILI/CLI sentinel sites could be routinely used as a surveillance tool for vaccine effectiveness monitoring.

## Figures and Tables

**Figure 1 vaccines-12-00954-f001:**
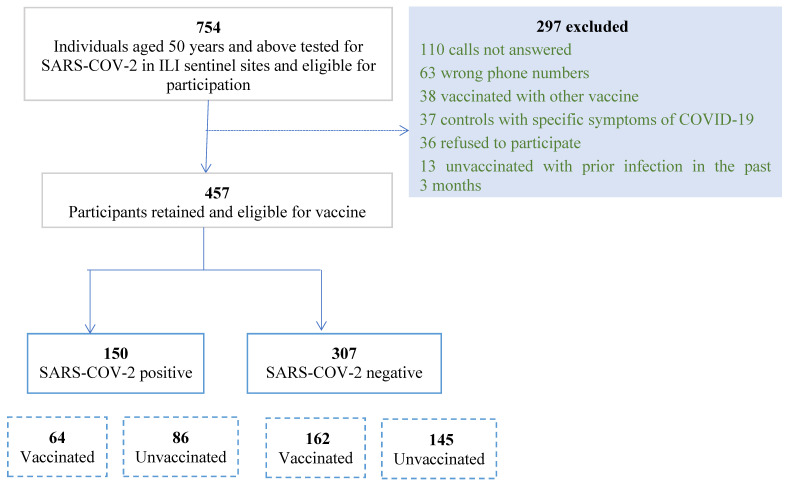
Flowchart of patient recruitment for test-negative case–control study on SARS-CoV-2 infection among ILI/CLI Patients, July–December 2021, Lebanon.

**Table 1 vaccines-12-00954-t001:** Characteristics of the study participants, July–December 2021, Lebanon (*n* = 457).

		Cases (*n* = 150)		Controls (*n* = 307)		
Characteristics	Options	Number	%	Number	%	*p*-Value
**Age groups**	50–60	84	56	184	61	0.501
	61–70	45	30	91	29	
	>70	21	14	32	10	
**Sex**	Male	66	44	124	40	0.462
	Female	84	56	183	60	
**Nationality**	Lebanese	100	70	235	77	0.117
	Non-Lebanese	44	30	72	23	
**Place of residence**	Beirut/ML	59	41	118	39	0.803
	Nabatieh/South	37	26	91	30	
	Akkar/North	35	24	73	24	
	Bekaa/BH	13	9	23	7	
**Smoking status**	Current smoker	82	57	120	39	0.001
	Noncurrent smoker	63	43	185	61	
**Health conditions in the previous 12 months**						
**Presence of comorbidities**	No comorbidities	61	41	115	37	0.5
	At least one comorbidity	89	59	192	63	
**GP consultation for any condition**	Yes	87	63	202	68	0.362
	No	52	37	93	32	
**Hospital admission for any conditions**	Yes	31	22	60	20	0.666
	No	109	78	235	80	
**Living conditions**						
**Covering basic needs**	Yes	72	50	136	45	0.314
	No	73	50	169	55	
**Main income source**	Work income	60	41	131	43	0.824
	Family help	58	40	109	36	
	Financial help	10	7	23	7	
	Others	17	13	42	14	

**Table 2 vaccines-12-00954-t002:** The vaccination status of the study participants, July–December 2021, Lebanon (*n* = 457).

		Cases (*n* = 150)		Controls (*n* = 307)		
Variables	Options	Number	%	Number	%	*p*-Value
COVID-19 vaccination status ^a^	Not vaccinated	86	57	145	47	0.127
	Partially vaccinated	12	8	29	10	
	Fully vaccinated	52	35	133	43	
For those vaccinated, Median delays of symptom onset in days (IQR)	From first vaccine dose	25 (4–47)		44.5 (29–70)		0.17
	From second vaccine dose	116 (42–156)		98 (50–136)		0.47
Adverse reactions following first dose	No side effect	40	54	84	41	0.102
	Minor side effect	0	0	115	57	
	Moderate side effect	34	46	4	2	
Adverse reactions following second dose	No side effect	36	52	69	40	0.18
	Minor side effect	34	48	103	59	
	Moderate side effect	0	0	2	1	

^a.^ Patients were considered unvaccinated if they did not receive a COVID-19 vaccine or if they were vaccinated on the same day as or after the onset of symptoms. Patients were considered partially vaccinated if they received one of two doses at least 14 days before the onset of symptoms or received two doses with the second dose received <14 days before the onset of symptoms. Patients were considered fully vaccinated if they received both doses at least 14 days before the onset of symptoms. IQR—interquartile range.

**Table 3 vaccines-12-00954-t003:** Unadjusted and adjusted vaccine effectiveness estimates of Pfizer COVID-19 mRNA vaccines against laboratory-confirmed symptomatic SARS-CoV-2 infection between July and December 2021 in Lebanon (*n* = 197).

	Cases		Controls					
COVID-19 Vaccination Status	Number	%	Number	%	Unadjusted OR [95% CI]	Unadjusted VE % [95% CI]	Adjusted OR [95% CI]	Adjusted VE % [95% CI]
Partially vaccinated	11	8.4	25	9.33	0.74 [0.34, 1.6]	26 [−60, 66]	0.78 [0.35, 1.7]	22 [−70, 65]
Fully vaccinated	45	34	116	43	0.65 [0.44, 1.02]	35 [−2, 56]	0.56 [0.33, 0.94]	44 [6, 67]

The model was adjusted for age, sex, nationality, comorbidities, socioeconomic status, exposure to SARS-CoV-2 in the community, and the practice of non-pharmaceutical interventions. Note that the dataset used for multivariable analysis, including the complete data on all variables, included 197 records.

## Data Availability

The data presented in this study are openly available in Harvard Dataverse: https://doi.org/10.7910/DVN/K7K5LB (accessed on 10 April 2024).

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
