# Peer review of "Pfizer-BioNTech (BNT162b2) Vaccine Effectiveness against Symptomatic Laboratory-Confirmed COVID-19 Infection among Outpatients in Sentinel Sites, Lebanon, July–December 2021"

_vaccines, 2024, doi:10.3390/vaccines12090954_

Round 1

Reviewer 1 Report

Comments and Suggestions for Authors

The authors described the efficacy of COVID-19 vaccination in Lebanon in 2021 among population over the age of 50. Although the study is old (2021), and as we all know that since then there occur many changes in the virus mutations (variants). The study is attractive but the following concerns should be addressed;

1-      Why the authors only focus on Pfizer-BioNTech vaccine, which is mRNA based? Does it mean the government of Lebanon only allowed this specific vaccine to be inoculated in the country or there is some other reason. Please specify.

2-      The author shall include some details about the Pfizer-BioNTech vaccines (regarding the mRNA vaccine against various variants of SARS-CoV-2). Please read:

https://doi.org/10.1038/d41586-021-02483-w

https://doi.org/10.1016/j.jconrel.2022.03.032

https://doi.org/10.1007/s12274-022-4627-5

3-      Please pay attention to the style of the references.

4-      There are grammatical errors and broken sentences. The whole manuscript shall be read be carefully.

5-      It should have been better to consider some disease condition in the age population  selected.

6-      On what basis the authors differentiated Minor and Major side effects? Please explain, and what were these symptoms?

Comments on the Quality of English Language

Careful reading of the whole manuscript is required. Thank you

Author Response

comment 1 : Why the authors only focus on Pfizer-BioNTech vaccine, which is mRNA based? Does it mean the government of Lebanon only allowed this specific vaccine to be inoculated in the country or there is some other reason. Please specify.

Response 1: Thank you for pointing this out. Actually, Pfizer-BioNTech vaccine was was one of the primary vaccines used for older adults in Lebanon including those aged 50 years and above. We added explanation in the background (line 39-42)

Comment 2 : The author shall include some details about the Pfizer-BioNTech vaccines (regarding the mRNA vaccine against various variants of SARS-CoV-2)

Response2: thank you for pointing this out. We have mentioned details on Pfizer-BioNTech vaccine effectiveness against alpha and delta variants in the discussion ( L 218-219) . To note that delta variant was the dominating one during the study period.

comment 3: Please pay attention to the style of the references.

Response 3: thank you for pointing this out ... We reviewed the style of cited references

Comment 4: There are grammatical errors and broken sentences. The whole manuscript shall be read be carefully.

Response4: thank you for your comment . We reviewed the manuscript and made the necessary modifications

Comment 5 : It should have been better to consider some disease condition in the age population  selected.

Response 5: We collected information on disease conditions in the selected age group but we didn't go for further analysis as we have a small sample size 

Comment 6:  On what basis the authors differentiated Minor and Major side effects? Please explain, and what were these symptoms?

Response 6: We differentiate between minor and major side effects after covid-19 vaccination. minor/moderate side effects such as injection site reaction, mild fever, fatigue, headache, etcc... that do not require medical attention or causing significant disruption to daily activities and often resolve within a few days. while, major side effects are those causing severe or serious reactions that may require medical attention and could potentially lead to hospitalization or long-term health concerns.

Reviewer 2 Report

Comments and Suggestions for Authors

In the manuscript titled “Pfizer-BioNTech (BNT162b2) vaccine effectiveness against

symptomatic laboratory-confirmed COVID-19 infection among outpatients in sentinel sites, Lebanon, July-December 2021”, the authors estimated the effectiveness of Pfizer BNT162b2 vaccine in preventing symptomatic laboratory-confirmed COVID-19. The authors investigated adults aged 50 years and older who presented with influenza like illness (ILI) or COVID-19 like illness (CLI) in surveillance sentinel sites between 1 July and 31 December 2021 via

test-negative case-control (TND) study. For the 457 participants with symptoms, the unvaccinated participants did not receive any vaccine, while the vaccinated participants received at least one dose within 14 days before onset of symptoms. They found 150 (33%) were positive and 307 (67%) were negative for SARS-CoV2. Consider vaccination status, the vaccine effectiveness was 22%for partially vaccinated and 44% for fully vaccinated. So, the conclusion is that two doses of Pfizer BioNTech vaccine were effective in preventing COVID-19 symptomatic illness in the older population. The paper is written well. The minor comments are as follows:

1.      Why the test negative case-control abbreviated as TND, not TNC?

2.      Table 1 and 2 don’t have headline.

Author Response

comment 1: Why the test negative case-control abbreviated as TND, not TNC?

Thank you for your comment. The abbreviation TND stands for test-negative design and we referred to WHO guidance on vaccine effectiveness studies for this abbreviation

comment 2 : Table 1 and 2 don’t have headline.

thank you for pointing this out : Table 1 and 2 have titles and we added options in table 1 second column to make it more clear. 

Reviewer 3 Report

Comments and Suggestions for Authors

In this research article, the authors studied retrospectively the efficacy of the Pfizer-BioNTech (BNT162b2) vaccine against the symptoms of SARS-CoV-2-positive patients, in Lebanon.

As defined also by the authors, the continuous monitoring of the efficacy and of safety of the vaccine is very important and it is a goal standard for the good practice, but the manuscript should be improved to allow a better comprehension of the data and of the results.

-       The authors underlined that they studied Lebanese population, and, in the introduction, they declared that after clinical trials is important to consider the efficacy of the vaccine in real life conditions, including regional differences in population characteristics (lines 49-50). In table1 they reported that not all the population is Lebanese. In my opinion, this condition should be considered as exclusion criteria.

-       In the material and methods section, the author should explain better the study design and define what they want to study.

-       Why having symptoms compatible with COVID-19 is it considered as exclusion criteria for control? Did not the author consider the population consulting at sentinel clinics with upper respiratory infection symptoms? The exclusion of these patients could introduce a bias in the statistical analysis. Maybe, another more useful exclusion criteria for the case population should have been the use of some pharmaceutical drugs that could interfere with the symptomatology.

-       The author reported that the onset of the symptoms was longer in control group. Considering that these patients are SARS-CoV-2 negative is obvious that it depends on the kind of pathogen that infected them. So, I do not think that is relevant, it is more interesting a consideration between vaccinated and unvaccinated SARS-CoV-2 positive patients. Moreover, this comparison it is important for all the considered parameter, considering the scopus of the study.

-       In line 155 and in table 2, the author declared that in case group, vaccinated persons are 64 (12 partially vaccinated and 52 fully vaccinated). Otherwise, in 3.5 paragraph, in case group, fully vaccinated persons are 45 and partially are 11. The author should explain this point.

-       Please, correct the typos and align columns in table 2.

-       Please, explicit VE in line 186

Author Response

comment1 : The authors underlined that they studied Lebanese population, and, in the introduction, they declared that after clinical trials is important to consider the efficacy of the vaccine in real life conditions, including regional differences in population characteristics (lines 49-50). In table1 they reported that not all the population is Lebanese. In my opinion, this condition should be considered as exclusion criteria.

Response 1: thank you for pointing this out . Actually we assessed the effectiveness of the vaccine in the population residing in Lebanon and therefore our participants who were visiting the outpatients sentinel sites were not only Lebanese and include other nationalities.

comment 2 :   In the material and methods section, the author should explain better the study design and define what they want to study.

Response 2: Thank you for your comment.. We will appreciate if you please indicate what is exactly is missing or not clear in the methods section or if you suggest any definitions to be added.

comment 3: Why having symptoms compatible with COVID-19 is it considered as exclusion criteria for control? Did not the author consider the population consulting at sentinel clinics with upper respiratory infection symptoms? The exclusion of these patients could introduce a bias in the statistical analysis. Maybe, another more useful exclusion criteria for the case population should have been the use of some pharmaceutical drugs that could interfere with the symptomatology.

Response 3: We excluded controls with specific symptoms for COVID-19 such as loss of smell and taste to eliminate laboratory testing bias as they can be false negative and we referred to WHO guidance on covid-19 vaccine effectiveness for this exclusion criteria... and for the second part of the comment, we agree with you it would be useful to consider it as exclusion criteria but unfortunately we don't have related data. 

comment 4:  The author reported that the onset of the symptoms was longer in control group. Considering that these patients are SARS-CoV-2 negative is obvious that it depends on the kind of pathogen that infected them. So, I do not think that is relevant, it is more interesting a consideration between vaccinated and unvaccinated SARS-CoV-2 positive patients. Moreover, this comparison it is important for all the considered parameter, considering the scopus of the study.

Response 4: thank you for your comment. We agreed with you that this is just an additional information indicating median delay for symptoms onset is higher among controls than among cases.  

comment 5 : In line 155 and in table 2, the author declared that in case group, vaccinated persons are 64 (12 partially vaccinated and 52 fully vaccinated). Otherwise, in 3.5 paragraph, in case group, fully vaccinated persons are 45 and partially are 11. The author should explain this point.

response 5: Thank for pointing this out. The difference is due to the logistic regression . After applying logistic regression to data set and taking into considerations the variables included in the model, all rows with missing data will be deleted which explain the indicated difference

comment 6:   Please, correct the typos and align columns in table 2.

Response 6: Thank you for your comment. Typo corrected and columns aligned 

Comment 7: Please, explicit VE in line 186

Response 7 : As suggested we made the necessary modifications in line 186

Reviewer 4 Report

Comments and Suggestions for Authors

On 31 December 2020, BNT162b2 was listed for emergency use by the World Health Organisation, representing the first COVID-19 vaccine to receive emergency validation via this pathway. At the same time, the vaccine still provides effective protection against variants of the virus, but its effectiveness against both severe and mild disease caused by the omicron variant after administration of two doses turned out to be lower compared to the delta variant, while the waning of protection occurs more rapidly. Thus, the topic of this article is still relevant today. Of course, randomized controlled trials are preferable, but case-control studies can also be useful material for systematic reviews. In this study, appropriate methods of population analysis were used, and the results are statistically sound. I have a generally positive assessment of the research conducted, but I have a few comments:

(1) In the "Introduction" section, the authors did not explain why they included only patients older than 50 years in the study.

(2) Section 2.6. Exposure assessment: the dose of the vaccine is not specified, i.e. whether only the standard dose of 30 mcg was used in all cases.

(3) Section 3. There were no cases of vaccination-related adverse events, including fever, fatigue, nausea, anaphylaxis, arthralgia, and others.

(4) Table 1. Among the concomitant diseases, it is advisable to indicate the presence of pathologies requiring immunosuppressive and hormonal therapy, as well as type 2 diabetes mellitus, if present.

(5) The "Discussion" section. It is advisable to provide a brief comparative description of the BNT162b2 vaccine in relation to the effectiveness of other vaccines recommended by the WHO.

(6) References must be adapted to the standard MDPI style.

Author Response

Comment 1:  In the "Introduction" section, the authors did not explain why they included only patients older than 50 years in the study.

Response 1: Thank you for pointing this out. We included only patients older than 50 years as this age group are exclusively receiving BNT162b2 vaccine according to the national Policy. we added an explanation in the introduction ( L39-L42)

comment 2:  Exposure assessment: the dose of the vaccine is not specified, i.e. whether only the standard dose of 30 mcg was used in all cases.

Response 2: Delivery of the vaccine was monitored by the public authority and therefore all health facilities were following the same protocol for vaccine delivery including standard dosage of 30 mcg.

Comment 3: There were no cases of vaccination-related adverse events, including fever, fatigue, nausea, anaphylaxis, arthralgia, and others.

Response3: thank you for your comment. minor and major Side effects after vaccination were presented in table 2 in results section

comment 4: Among the concomitant diseases, it is advisable to indicate the presence of pathologies requiring immunosuppressive and hormonal therapy, as well as type 2 diabetes mellitus, if present.

Response4: Thank you for pointing this out... we considered to collect information on participants comorbidities but we didn't go for further analysis as we have a small sample size and therefore it will be difficult to obtain significant results

comment 5: The "Discussion" section. It is advisable to provide a brief comparative description of the BNT162b2 vaccine in relation to the effectiveness of other vaccines recommended by the WHO.

Response 5: We agreed with you on the importance of this comparison but our study was only focusing on BNT162b2 vaccine and we found this to be out of the study scope

comment 6:  References must be adapted to the standard MDPI style.

Response 6: thank you for your comment ... We adapted references to MDPI style.